# Transferring Hierarchical Structure with Dual Meta Imitation Learning

**Chongkai Gao**
Department of Automation
Tsinghua University
gck20@mails.tsinghua.edu.cn

**Yizhou Jiang**
Department of Automation
Tsinghua University
jyz20@mails.tsinghua.edu.cn

**Feng Chen**
Department of Automation, Tsinghua University
LSBDPA Beijing Key Laboratory
chenfeng@mail.tsinghua.edu.cn

**Abstract:** Hierarchical Imitation Learning (HIL) is an effective way for robots to acquire sub-skills from long-horizon unsegmented demonstrations. However, the learned hierarchical structure lacks the mechanism to transfer across multi-tasks or adapt to new tasks, which makes them have to learn from scratch when facing a new situation. Transferring and reorganizing modular sub-skills require fast adaptation abilities of both the high-level network and sub-skills to reschedule new forms of sub-skills in new tasks. In this work, we propose Dual Meta Imitation Learning (DMIL), a hierarchical meta imitation learning method where the high-level network and sub-skills are iteratively meta-learned with model-agnostic meta-learning (MAML [1]). DMIL uses the likelihood of state-action pairs from each sub-skill as the supervision for the high-level network adaptation, and uses the adapted high-level network to determine different data set for each sub-skill adaptation. We theoretically prove the convergence of the iterative training process of DMIL and establish the connection between DMIL and Expectation-Maximization algorithm. Empirically, we achieve state-of-the-art few-shot imitation learning performance on the Meta-world [2] benchmark and competitive results on long-horizon tasks of Kitchen environments.

**Keywords:** Hierarchical Imitation Learning, Meta Learning

## 1 Introduction

Imitation learning (IL) has shown promising results for intelligent robots to conveniently acquire skills from expert demonstrations [3, 4]. Nevertheless, imitating long-horizon unsegmented demonstrations has been a challenge for IL algorithms, because of the well-known issue of compounding errors [5]. This is one of the crucial problems for the application of IL methods to robots since plenty of practical manipulation tasks are long-horizon. Hierarchical Imitation Learning (HIL) aims to tackle this problem by decomposing long-horizon tasks with a hierarchical model, in which a set of sub-skills are employed to accomplish specific parts of the long-horizon task, and a high-level network is responsible for determining the switching of sub-skills along with the task. Such a hierarchical structure is usually modeled with Options trained with Expectation-Maximization algorithm [6, 7, 8] or goal-conditioned IL paradigms [9]. HIL expresses the nature of how humans solve complex tasks, and has been considered to be a valuable direction for IL algorithms [10].

However, most current HIL methods have no explicit mechanism to transfer previously learned sub-skills to new tasks with few-shot demonstrations. This requirement comes from that the learned hierarchical structure may conflict with discrepant situations in new tasks. As shown in fig. 1(a), both the high-level network and sub-skills need to be transferred to new forms to satisfy new requirements: the high-level network needs new manners to schedule sub-skills in new tasks (for example, calling different sub-skills *pulling* or *pushing* at the same state), and each sub-skill needs to adapt to new specific forms in new tasks (for example, grasping different kinds of objects). This drives us to develop new methods to endow HIL with the ability to simultaneously transfer both the high-level network and sub-skills with few-shot new task demonstrations.

6th Conference on Robot Learning (CoRL 2022), Auckland, New Zealand.

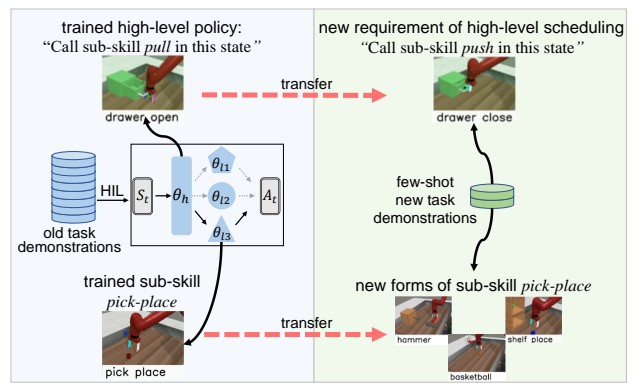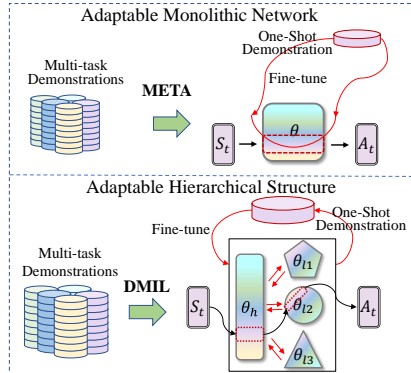

(a) Illustration of the bi-level transfer problem of HIL in new tasks.   (b) Comparison of MIL and DMIL.

Figure 1: (a) Both the high-level network and sub-skills need to be transferred to new tasks. Above: when the robot arm is over a half-open drawer, the task can be either opening or closing the drawer, which requires the high-level network to call different sub-skills. Below: the same sub-skill *pick-place* may exhibit different specific forms in new tasks. (b) DMIL aims to integrate MAML into HIL with a novel iterative optimization procedure that meta-learns both the high-level network and sub-skills.

Recently, meta imitation learning (MIL) [11, 12, 13] employs model-agnostic meta-learning (MAML) [1] into the imitation learning procedure to enable the learned policy to quickly adapt to new tasks with few shot demonstrations. MAML first fine-tunes the policy network in the inner loop, then evaluates the fine-tuned network to update its initial parameters with end-to-end gradient descent at the outer loop. The success of MIL inspires us to integrate MAML into HIL to transfer the hierarchical structure in new tasks. However, this is not straightforward. HIL is a bi-level structure that is trained in an iterative and self-supervised paradigm [6], thus both the high-level network and sub-skills need to be meta-learned by MAML. Intuitively, the question is: should sub-skills choose the fine-tuned high-level network or the original high-level network for their inner loops and outer loops? The same question applies to the high-level network. This is a dual meta learning problem, and we need to explore appropriate methods for MAML to schedule the fine-tune steps (inner loops) and meta-update steps (outer loops) of the bi-level network in HIL to ensure convergence.

In this work, we propose a novel hierarchical meta imitation learning framework called Dual Meta Imitation Learning (DMIL) to incorporate MAML into the iterative training process of HIL, as shown in fig. 1(b). Based on the common EM-like HIL structure [6, 7] we design an elaborate bi-level MAML procedure for this hierarchical structure to make it can be fully meta-learned. In this procedure, we first fine-tune the high-level network and sub-skills *in sequence* at inner loops, then meta-update them *simultaneously* at outer loops. We theoretically prove the convergence of this special training procedure by leveraging previous results from [14, 15, 16] to reframe both MAML and DMIL as hierarchical Bayes inference processes and get the convergence of DMIL according to the convergence of MAML from previous results [17]. We test our method on the challenging meta-world benchmark environments [2] and the Kitchen environment of D4RL benchmarks [18]. In our experiments, we successfully acquire a set of meaningful sub-skills from a large scale of manipulation tasks, and achieve state-of-the-art few-shot imitation learning abilities in the ML45 suite. In summary, the main contributions of this paper are:

- We propose DMIL, a novel hierarchical meta imitation learning framework that meta-learns both the high-level network and sub-skills from unsegmented multi-task demonstrations in a general EM-like fashion, and theoretically prove its convergence.
- We achieve state-of-the-art few-shot imitation learning performance on meta-world benchmark environments and competitive results in the Kitchen environment.

## 2 Related Work

### 2.1 Hierarchical Imitation Learning

Recovering inherent sub-skills contained in expert demonstrations and reusing them with hierarchical structures has long been an interesting topic in the hierarchical imitation learning (HIL) domain. According to whether there are pretraining tasks, we can divide HIL methods into two categories. The first one employs a set of manually designed pretraining tasks that encourage distinct skills

or primitives, then learn a high-level network to master the switching of primitives to accomplish complex tasks [19, 20, 21, 22, 9]. However, for unsegmented demonstrations where no pretraining tasks are provided, which is more common in the real world, these methods can not be applied.

The second kind of methods aim to learn sub-skills with unsupervised learning methods. Daniel et al. [6], Krishnan et al. [7] acquire Options [23] from demonstrations with an Expectation-Maximization-like procedure and use the Baum-Welch algorithm to estimate the parameters of different options. Henderson et al. [24], Jing et al. [8] integrate generative adversarial networks into the option discovery process. Li et al. [25], Sharma et al. [26], Lee and Seo [27] incorporate generative-adversarial imitation learning [28] framework and an information-theoretic metric [29] to simultaneously imitate the expert and maximize the mutual-information between latent sub-skill categories and corresponding trajectories to acquire decoupled sub-skills. There are also some methods called mixture-of-expert (MoE) that compute the weighted sum of all primitives to get the action rather than only using one of them at each time step [30, 31, 32]. Other methods aim to seek an appropriate latent space that can map sub-skills into it and then condition a policy on the latent variable generated from the latent space to reuse sub-skills [33, 34, 35, 36, 37]. In this work, we use the EM-like HIL methods used in [6, 7] as they can acquire semantic and separated sub-skill networks which have shown better results in recent works [38], and compare other kinds of HIL methods as baselines to show the superiority of our method in experiments. None of above HIL methods do not take the fast adaptation ability into consideration: they assume testing tasks are in the same distribution of training tasks. Although some work fine-tune the whole structure in new tasks [36], the performance of fine-tuning all depends on the generalization of deep networks, which may vary among different tasks and network designs. Thus we aim to enhance HIL methods with meta learning abilities, as described below.

## 2.2 Meta Imitation Learning

Meta imitation learning, or one-shot imitation learning, leverages various meta-learning methods and multi-task demonstrations to meta-learn a policy that can be quickly adapted to a new task with few-shot new task demonstrations. Duan et al. [39], Cachet et al. [40] employ self-attention modules to process the whole demonstration and the current observation to predict the current action. Yu et al. [13], Finn et al. [12], Yu et al. [11] use model-agnostic meta-learning (MAML) [1] to achieve one-shot imitation learning ability for various manipulation tasks with robot or human visual demonstrations. Xu et al. [41], Yu et al. [42] propose to meta-learn a robust reward function that can be quickly adapted to new tasks and then use it to perform IRL in new tasks. However, they need downstream inverse reinforcement learning after the adaptation of reward functions, thus conflicts with our goal of few-shot adaptation. Most above methods only learn one monolithic policy, lacking the ability to model multiple sub-skills in long-horizon tasks. Some works aim to tackle the multi-modal data problem in meta-learning by avoiding single parameters initialization across all tasks [43, 44, 45, 46], but they lack the mechanism to schedule the switching of different sub-skills over time. There are some works that also meta-learn a set of sub-skills in a hierarchical structure [13, 45], but they either use manually designed pretraining tasks or relearn the high-level network in new tasks, which is not appropriate in few-shot imitation learning settings.

## 3 Method

### 3.1 Meta Imitation Learning

We denote a discrete-time finite-horizon Markov decision process (MDP) as a tuple $(\mathcal{S}, \mathcal{A}, T, P, r, \rho_0)$, where $\mathcal{S}$ is the state space, $\mathcal{A}$ is the action space, $T$ is the time horizon, $P : \mathcal{S} \times \mathcal{A} \times \mathcal{S} \rightarrow [0, 1]$ is the transition probability distribution, $r : \mathcal{S} \times \mathcal{A} \rightarrow \mathbb{R}$ is the reward function, and $\rho_0$ is the distribution of the initial state $s_0$. The goal of meta imitation learning is to extract some common knowledge from a set of robot manipulation tasks $\{\mathcal{T}_i\}$ that come from the same task distribution $p(\mathcal{T})$, and adapt it to new tasks quickly with few shot new task demonstrations. As in model-agnostic meta-learning algorithm (MAML) [1], we formalize the common knowledge as the initial parameter $\theta$ of the policy network $\pi_\theta$ that can be efficiently adapted with new task gradients.

For each task $\mathcal{T}_i \sim p(\mathcal{T})$, a set of demonstrations $\mathcal{D}_i$ is provided, where $\mathcal{D}_i$ consists of $N$ demonstration trajectories: $\mathcal{D}_i = \{\tau_{ij}\}_{j=1}^N$, and $\tau_{ij}$ consists of a sequence of state-action pairs: $\tau_{ij} = \{(s_t, a_t)\}_{t=1}^{T_{ij}}$, where $T_{ij}$ is the length of $\tau_{ij}$. Each $\mathcal{D}_i$ is randomly split into support set $\mathcal{D}_i^{tr}$ and query set $\mathcal{D}_i^{val}$ for meta-training and meta-testing respectively. During the training phase, we sample $m$ tasks from $p(\mathcal{T})$, and in each task $\mathcal{T}_i$, we use $\mathcal{D}_i^{tr}$ to fine-tune $\pi_\theta$ to get the adapted task-specific parameters $\lambda_i$ with gradient descent, and then evaluate it with $\mathcal{D}_i^{val}$ to get the meta-gradient of $\mathcal{T}_i$, and we optimize the initial parameters $\theta$ with the average of meta-gradients from all $m$ tasks. The policy

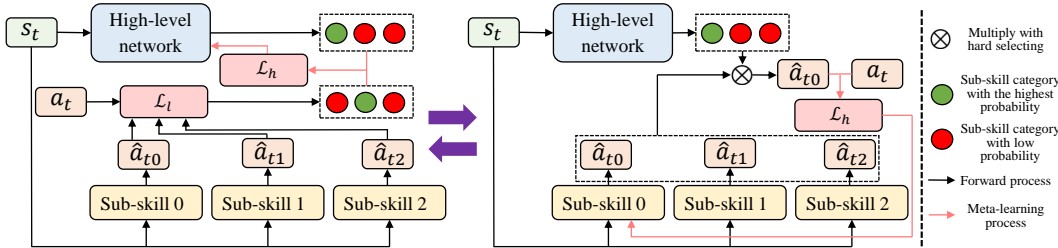

Figure 2: The iterative meta-learning process of DMIL at each iteration. Left: the supervision of high-level network (sub-skill categories) comes from the most accurate sub-skill (the green one, sub-skill 1 here). Right: the sub-skill updated at current step (the green one, sub-skill 0 here) is determined by the fine-tuned high-level network.

$\pi_\theta : \mathcal{S} \to \mathcal{A}$ is trained to maximize the likelihood such that $\theta^* = \arg\max_\theta \sum_{i=1}^N \log \pi_\theta(a_i|s_i)$, where $N$ is the number of provided state-action pairs. We denote the loss function of this optimization problem as $\mathcal{L}_{BC}(\theta, \mathcal{D})$, and the general objective of meta imitation learning problem is:

$$\min_\theta \sum_{i=1}^m \mathcal{L}_{BC}\left(\lambda_i, \mathcal{D}_i^{\text{val}}\right), \tag{1}$$

where $\lambda_i = \theta - \alpha\nabla_\theta\mathcal{L}_{BC}(\theta, \mathcal{D}_i^{tr})$, and $\alpha$ is a hyper-parameter which represents the inner-update learning rate.

### 3.2 Dual Meta Imitation Learning (DMIL)

In this work we assume at each time step $t$, the robot may switch to different sub-skills to accomplish the task. We define the sub-skill category at each time step $t$ as $z_t = 1, \cdots, K$, where $K$ is the maximum number of sub-skills. We assume a successful trajectory $\tau_{ij}$ of a task $\mathcal{T}_i$ is generated from several (at least one) sub-skill policies, i.e., $\tau_{ij} = \sum_{t=1}^{T_{ij}}\{(s_t, \pi_E(s_t|z_t))\}$, where $\pi_E$ represents the expert policy. Our goal is to learn a hierarchical structure from multi-task demonstrations $\{\mathcal{D}_1, \cdots, \mathcal{D}_m\}$ in an unsupervised fashion. The high-level network $\pi_{\theta_h}$ that parameterized by $\theta_h$ determines the sub-skill category $\hat{z}_t$ at each time step $t$, and the $z$-th sub-skill among $K$ different sub-skills $\pi_{\theta_{l1}}, \cdots, \pi_{\theta_{lK}}$ will be called to predict the corresponding action $\hat{a}_t$ of state $s_t$, where the hat symbol denotes the predicted result. We use $\lambda_h$ and $\lambda_{l1}, \cdots, \lambda_{lk}$ to represent the adapted parameters of $\theta_h$ and $\theta_{l1}, \cdots, \theta_{lK}$ respectively. We condition the high-level network only on states, i.e., $\hat{z}_t = \pi_{\theta_h}(s_t)$, to fit the actual situation at the testing phase.

DMIL aims to first fine-tune both $\pi_{\theta_h}$ and $\pi_{\theta_{l0}}, \cdots, \pi_{\theta_{lK}}$ and then meta-update them. In a new task, $\pi_{\theta_h}$ may not provide correct sub-skill categories as stated in the introduction. However, sub-skills still retain the ability to give out supervision for the high-level network with knowledge learned from previously learned tasks and few-shot demonstrations. This is because most robot manipulation tasks are made up of a set of shared basis skills like *reach, push* and *pick-place*. As shown on the left side of fig. 2, the sub-skill that gives out the closet $\hat{a}_t$ to $a_t$ can be seen as supervision for the high-level network to classify $s_t$ into this sub-skill. On the other hand, the adapted high-level network can classify each data point in provided demonstrations to different sub-skills for them to perform fine-tuning, as shown on the right side of fig. 2. In summary, DMIL contains four steps for one training iteration. We call them **High-Inner-Update (HI)**, **Low-Inner-Update (LI)**, **High-Outer-Update (HO)**, and **Low-Outer-Update (LO)**, which represents the fine-tuning and meta-updating process of the bi-level networks respectively. The key problem is how to arrange these optimization orders to ensure convergence. We first introduce these steps formally here, then discuss why they can achieve convergence in the next section. The whole procedure is summarized in algorithm 1 in appendix.

**HI:** For each sampled task $\mathcal{T}_i$, we sample the first batch of trajectories $\{\tau_{i1}\}$ from $\mathcal{D}_i^{tr}$. The principle of this step is to use sub-skill that can predict the closest action to the expert action to provide self-supervised category ground truths for the training of high-level network, which is a classifier in form. We make *every* state-action pair passed directly to each sub-skill and compute $\mathcal{L}_{BC}(\theta_{lk}, \tau_{i1}), k = 1, \cdots, K$, and choose the ground truth at each time step as the sub-skill category $k$ that minimizes $\mathcal{L}_{BC}(\theta_{lk}, (s_t, a_t))$:

$$p(z_{i1t} = k) = \begin{cases} 1, & \text{if } k = \arg\min_k \mathcal{L}_{BC}(\theta_{lk}, (s_t, a_t)) \\ 0, & \text{else} \end{cases}. \tag{2}$$

Then we get predicted sub-skill categories from the high-level network: $\hat{z_{i1t}} = \pi_{\theta_h}(s_t)$, and use a cross-entropy loss to train the high-level network:

$$\mathcal{L}_h(\theta_h, \tau_{i1}) = -\frac{1}{T_{i1}} \sum_{t=1}^{T_{i1}} \sum_{k=1}^{K} p(z_{i1t} = k) \log p(\hat{z_{i1t}} = k). \tag{3}$$

Finally we perform gradient descent on the high-level network and get $\lambda_h = \theta_h - \alpha \nabla_{\theta_h} \mathcal{L}_h(\theta_h, \tau_{i1})$. Note $\theta_{l1}, \cdots, \theta_{lk}$ are freezed here.

**LI:** We sample the second batch of trajectories $\{\tau_{i2}\}$ from $\mathcal{D}_i^{tr}$. The adapted high level network $\pi_{\lambda_h}$ will process each state in $\tau_{i2}$ to get sub-skill category $\hat{z_{i2t}} = \pi_{\lambda_h}(s_t)$ at each time step, thus we get $K$ separate data sets for different sub-skills: $\mathcal{D}_{2k} = \{(s_{i2t}, a_{i2t}) | \hat{z_{i2t}} = k\}, k = 1, \cdots, K$. Then we compute the adaptation loss for each sub-skill with the corresponding dataset. In case of continuous action space, we assume that actions belong to Gaussian distributions, so we have:

$$\mathcal{L}_{BC}(\theta_{lk}, \mathcal{D}_{2k}) = -\frac{1}{N_k} \sum_{t=1}^{N_k} (a_t - \pi_{\theta_{lk}}(s_t))^2, \tag{4}$$

where $N_k$ is the number of state-action pairs in $\mathcal{D}_{2k}$. Finally we perform gradient descent on sub-skills and get $\lambda_{lk} = \theta_{lk} - \alpha \nabla_{\theta_{lk}} \mathcal{L}_{BC}(\theta_{lk}, \mathcal{D}_{2k})$. Note $\pi_{\lambda_h}$ is frozen in this process.

**HO:** We sample the third batch of trajectories $\{\tau_{i3}\}$ from $\mathcal{D}_i^{tr}$ and get $\mathcal{L}(\lambda_h, \tau_{i3})$ as in the HI process. Then we use it to compute the meta-gradient $\nabla_{\theta_h} \mathcal{L}(\lambda_h, \tau_{i3})$ which equals to:

$$\nabla_{\lambda_h} \mathcal{L}(\lambda_h, \tau_{i3})|_{\lambda_h = \theta_h - \alpha \nabla_{\theta_h} \mathcal{L}(\theta_h, \tau_{i2})} * \nabla_{\theta_h} \lambda_h. \tag{5}$$

**LO:** We sample $\tau_{i4}$ and get $\mathcal{L}(\lambda_{lk}, \mathcal{D}_{4k})$, $k = 1, \cdots, K$ as in the LI process, then we use it to compute the meta-gradient $\nabla_{\theta_{lk}} \mathcal{L}(\lambda_{lk}, \mathcal{D}_{4k})$ which equals to:

$$\nabla_{\lambda_{lk}} \mathcal{L}(\lambda_{lk}, \mathcal{D}_{4k})|_{\lambda_{lk} = \theta_{lk} - \alpha \nabla_{\theta_{lk}} \mathcal{L}(\theta_{lk}, \mathcal{D}_{1k})} * \nabla_{\theta_{lk}} \lambda_{lk}. \tag{6}$$

Note after the training of $m$ tasks, we average all meta-gradients from $m$ tasks and perform gradient descents on the initial parameters *together* to update high-level parameters $\theta_h' = \theta_h - \beta \sum_{i=1}^{m} \nabla_{\theta_h} \mathcal{L}(\lambda_h, \tau_{i3})$ and sub-skill policies parameters $\theta_{lk}' = \theta_{lk} - \beta \sum_{i=1}^{m} \nabla_{\theta_{lk}} \mathcal{L}(\lambda_{lk}, \tau_{i4})$, $k = 1, \cdots, K$, i.e., we do not update them at step 5 and 6. This is crucial to ensure convergence.

For testing, although our method needs two batches of trajectories for one round of adaptation, in practice we find only using one trajectory to perform HI and LI also works well in new tasks, thus DMIL can satisfy the one-shot imitation learning requirement. Besides the above process, we also add an auxiliary loss to better drive out meaningful sub-skills to avoid the excessively frequently switching between different sub-skills along with time. Detailed information can be found in C.

## 4 Theoretical Analysis

In this section, we show the above algorithm can converge by rewriting both MAML and DMIL as hierarchical variational Bayes problems to establish the equivalence between them since the convergence of MAML can be proved in Fallah et al. [17]. Proofs of all theorems are in Appendix D.

### 4.1 Hierarchical Variational Bayes Formulation of MAML

According to [14], MAML is a hierarchical variational Bayes inference process. The general meta-learning objective (1), which can be rewritten as $\mathcal{L}_g = \mathcal{L}(\theta, \lambda_1, \cdots, \lambda_m) = \log \prod_{i=1}^{m} p(\mathcal{D}_i | \theta)$, can be formulated as follows:

$$\mathcal{L}_g \geq \sum_{i=1}^{m} \{ \text{KL}(q(\phi_i; \lambda_i) \| p(\phi_i | \mathcal{D}_i, \theta) + E_{q(\phi_i; \lambda_i)}[\log p(\mathcal{D}_i, \phi_i | \theta) - \log q(\phi_i; \lambda_i)] \}, \tag{7}$$

where $\phi_i, i = 1, \cdots, m$ represent the local latent variables for task $\mathcal{T}_i$, and $\lambda_1, \cdots, \lambda_M$ are the variational parameters of the approximate posteriors over $\phi_1, \cdots, \phi_M$. We denote $\lambda_i$ as $\lambda_i(\mathcal{D}_i, \theta)$ and $p(\phi_i | \mathcal{D}_i, \theta)$ as $p(\phi_i | \mathcal{D}_i^{tr}, \theta)$ to mean that $\lambda_i$ and $\phi_i$ are determined with prior parameters $\theta$ and support data $\mathcal{D}_i^{tr}$. First we need to minimize $\text{KL}(q(\phi_i; \lambda_i) \| p(\phi_i | \mathcal{D}_i^{tr}, \theta))$ w.r.t. $\lambda_i$. According to D.2, we have:

$$\lambda_i(\mathcal{D}_i^{tr}, \theta) = \arg\max_{\lambda_i} E_{q(\phi_i; \lambda_i)}[\log p(\mathcal{D}_i^{tr} | \phi_i)] - \text{KL}(q(\phi_i; \lambda_i) \| p(\phi_i | \theta)), \tag{8}$$

We can establish the connection between 8 and the inner loop in MAML by the following Lemma:

**Lemma 1** In case $q(\phi_i; \lambda_i)$ is a Dirac-delta function and choosing Gaussian prior for $p(\phi_i | \theta)$, equation 8 equals to the inner-update step of MAML, that is, maximizing $\log p(\mathcal{D}_i^{tr})$ w.r.t. $\lambda_i$ by early-stopping gradient-ascent with choosing $\mu_\theta$ as initial point:

$$\lambda_i(\mathcal{D}_i^{tr}; \theta) = \mu_\theta + \alpha \nabla_\theta \log p(\mathcal{D}_i^{tr} | \theta)|_{\theta = \mu_\theta}. \tag{9}$$

Then we need to optimize $\mathcal{L}(\theta, \lambda_1, \cdots, \lambda_M)$ w.r.t. $\theta$. Since we evaluate $p(\mathcal{D}_i|\lambda_i(\mathcal{D}_i^{tr}, \theta))$ with only $\mathcal{D}_i^{val}$, we assume $p(\mathcal{D}_i|\lambda_i(\mathcal{D}_i^{tr}, \theta)) = p(\mathcal{D}_i^{val}|\lambda_i(\mathcal{D}_i^{tr}, \theta))$. We give out the following theorem to establish the connection between the meta-update process and the optimization of $\mathcal{L}_g$:

**Theorem 1** In case that $\Sigma_\theta \to 0^+$, i.e., the uncertainty in the global latent variables $\theta$ is small, the following equation holds:

$$\nabla_\theta \mathcal{L}_g = \sum_{i=1}^M \nabla_{\lambda_i} \log p(\mathcal{D}_i^{val}|\lambda_i) \nabla_\theta \lambda_i(\mathcal{D}_i^{tr}, \theta). \tag{10}$$

A general EM algorithm will first compute the distribution of latent variables (E-step), then optimize the joint distribution of latent variables and trainable parameters (M-step), and the likelihood of data can be proved to be monotone increasing to guarantee the convergence since the evidence lower bound of likelihood is monotone increasing. Here $\phi_i, i = 1, \cdots, M$ are the latent variables, and $\theta$ corresponds to the trainable parameters. Lemma 1 and Theorem 1 correspond to the E-step and M-step respectively. In the following part we establish the equivalence between 9 with 3 and 4, and between 10 with 5 and 6 to prove the equivalence between DMIL and MAML.

### 4.2 Modeling DMIL with Hierarchical Variational Bayes Framework

For simplicity, here we only derive in one specific task $\mathcal{T}_i$, since derivatives of parameters from multi-task can directly add up. We first establish the connection between the maximization of $\log p(\mathcal{D}_i^{tr}|\theta_h, \theta_{l1}, \cdots, \theta_{lK})$ with the particular loss functions in DMIL:

**Theorem 2** In case of $p(a_t|s_t, \theta_{lk}) \sim \mathcal{N}(\mu_{\theta_{lk}(s_t)}, \sigma^2)$, we have:

$$\nabla_{\theta_h} \log p(\mathcal{D}_i^{tr}|\theta_h, \theta_{l1}, \cdots, \theta_{lK}) = \nabla_{\theta_h} \mathcal{L}_h(\theta_h, \mathcal{D}_i^{tr}), \tag{11}$$

and

$$\nabla_{\theta_{lk}} \log p(\mathcal{D}_i^{tr}|\theta_h, \theta_{l1}, \cdots, \theta_{lK}) = \nabla_{\theta_{lk}} \mathcal{L}_{BC}(\theta_{lk}, \mathcal{D}_{2k}), \tag{12}$$

where $k = 1, \cdots, K$. Note in 12, $\mathcal{D}_{2k}$ corresponds to data sets determined by the adapted high level network $\lambda_h$, and this connects with 3 and 4 in DMIL. According to 8, finding $\lambda_i$ equals to maximize $\log p(\mathcal{D}_i^{tr}|\theta)$ in specific conditions, and here in Theorem 2, we prove that maximize $\log p(\mathcal{D}_i^{tr}|\theta_h, \theta_{l1}, \cdots, \theta_{lK})$ corresponds to 3 and 4 in DMIL. Thus theorem 2 corresponds to the E-step of DMIL, where we take $\tau_{i1}$ and $\tau_{i2}$ as $\mathcal{D}_i^{tr}$, and optimize $\arg\max_{\lambda_i} E_{q(\phi_i; \lambda_i)}[\log p(\mathcal{D}_i^{tr}|\phi_i)] - \mathrm{KL}(q(\phi_i; \lambda_i)\|p(\phi_i|\theta))$ with coordinate descent method, which can be proved to be equal to 9 in D.5.

For the M-step, we take $\tau_{i3}$ and $\tau_{i4}$ as $\mathcal{D}_i^{val}$. According to Theorem 1, we can take the derivative of $\lambda_{ih}, \lambda_{il1}, \cdots, \lambda_{ilK}$ to maximize the joint distribution of latent variables and trainable parameters to maximize the likelihood of dataset, so we have:

$$\nabla_{\theta_h, \theta_l} \log p(\mathcal{D}_i^{val}|\lambda_{ih}, \lambda_{il}) = [\nabla_{\lambda_{ih}} \log p(\mathcal{D}_i^{val}|\lambda_{ih}) * \nabla_{\theta_h} \lambda_{ih}(\mathcal{D}_i^{tr}, \theta_h), \\ \nabla_{\lambda_{il}} \log p(\mathcal{D}_i^{val}|\lambda_{il}) * \nabla_{\theta_l} \lambda_{il}(\mathcal{D}_i^{tr}, \theta_l)]^T \tag{13}$$

where $\theta_{il} = [\theta_{i1}, \cdots, \theta_{iK}]^T$ and $\lambda_{il} = [\lambda_{i1}, \cdots, \lambda_{iK}]^T$. This is exactly the gradients computed in HO and LO steps. Note this computation process can be automatically accomplished with standard deep learning libraries such as PyTorch [51]. To this end, we establish the equivalence between DMIL and MAML, and the convergence of DMIL can be proved.

For a clearer comparison, MAML is an iterative process of $\theta \to \lambda \to \theta'$, and DMIL is an iterative process of $\theta_h, \theta_l \to \lambda_h, \theta_l \to \lambda_h, \lambda_l \to \theta_h', \theta_l'$, where the posterior estimation stages $\theta_h, \theta_l \to \lambda_h, \theta_l \to \lambda_h, \lambda_l$ has no effect on parameters $\theta_h, \theta_l$, thus can be divided to two steps as in DMIL. This decoupled fine-tuning fashion is exactly what we need to first adapt the high-level network and then adapt sub-skills. If we end-to-end fine-tune parameters like $\theta_h, \theta_l \to \lambda_h, \lambda_l$, sub-skills will receive supervisions from an unadapted high-level network, which may provide incorrect classifications. Different to this, the meta-updating process $\lambda_h, \lambda_l \to \theta_h', \theta_l'$ must be done at the same time, since if we update $\theta_h$ and $\theta_l$ successively, the later one will receive different derivative (for example, $\nabla_{\theta_l} \log p(\mathcal{D}_i^{val}|\theta_{ih}', \lambda_{il})$) from derivatives in MAML ($\nabla_{\theta_l} \log p(\mathcal{D}_i^{val}|\lambda_{ih}, \lambda_{il})$), and the equivalence would not be proved.

## 5 Experiments

In experiments we aim to answer the following questions: (a) Can DMIL successfully transfer the learned hierarchical structure to new tasks with few-shot new task demonstrations? (b) Can DMIL achieve higher performance compared to other few-shot imitation learning methods? (c) What are the effects of different parts in DMIL, such as the skill number $K$, the bi-level meta-learning procedure, and the continuity constraint? Codes and video results are provided in supplementary materials.

Table 1: Success rates of different methods on Meta-world environments with $K = 3$. Each data point comes from 20 random seeds.

| | ML10 | | | | ML45 | | | |
|---|---|---|---|---|---|---|---|---|
| | Meta-training | | Meta-testing | | Meta-training | | Meta-testing | |
| Methods | 1-shot | 3-shot | 1-shot | 3-shot | 1-shot | 3-shot | 1-shot | 3-shot |
| OptionGAIL | 0.455±0.011 | **0.952±0.016** | 0.241±0.042 | 0.640±0.025 | 0.506±0.008 | 0.715±0.006 | 0.220±0.013 | 0.481±0.010 |
| MIL | **0.776±0.025** | 0.869±0.029 | 0.361±0.040 | 0.689±0.032 | 0.584±0.011 | 0.745±0.017 | 0.205±0.024 | 0.510±0.005 |
| PEMIRL | 0.598±0.023 | 0.810±0.007 | 0.162±0.003 | 0.256±0.009 | 0.289±0.051 | 0.396±0.024 | 0.105±0.005 | 0.126±0.008 |
| MLSH | 0.506±0.134 | 0.725±0.021 | 0.106±0.032 | 0.135±0.009 | 0.235±0.093 | 0.295±0.021 | 0.050±0.000 | 0.050±0.000 |
| DMIL | 0.775±0.010 | 0.949±0.009 | **0.396±0.016** | **0.710±0.021** | **0.590±0.010** | **0.859±0.008** | **0.376±0.004** | **0.640±0.009** |

Table 2: Cumulative rewards of different methods on four unseen tasks in Kitchen environment with $K = 4$. Boldface indicates excluded objects during training.

| Task (Unseen) | FIST-no-FT | SPiRL | DMIL(ours) |
|---|---|---|---|
| Microwave, Kettle, **Top Burner**, Light Switch | 2.0 ± 0.0 | **2.1 ± 0.48** | 1.5±0.48 |
| **Microwave**, Bottom Burner, Light Switch, Slide Cabinet | 0.0 ± 0.0 | 2.3 ± 0.49 | **2.35±0.39** |
| Microwave, **Kettle**, Hinge Cabinet, Slide Cabinet | 1.0 ± 0.0 | 1.9 ± 0.29 | **3.15±0.22** |
| Microwave, Kettle, Hinge Cabinet, **Slide Cabinet** | 2.0 ± 0.0 | **3.3 ± 0.38** | 2.95±0.44 |

## 5.1 Environments and Baselines

We choose to evaluate DMIL on two representative robot manipulation environments. The first one is Meta-world benchmark environments [2], which contains 50 diverse robot manipulation tasks, as shown in fig. 6 and fig. 7. We use both the ML10 suite (10 meta-training tasks and 5 meta-testing tasks) and ML45 suite (45 meta-training tasks and 5 meta-testing tasks) to evaluate our method, and collect 2K demonstrations for each task. We choose Meta-world since we think a large scale of diverse manipulation tasks can drive semantic skills. We use the following approaches for comparison in this environment: **Option-GAIL**: a hierarchical generative adversarial imitation learning method to discover options from unsegmented demonstrations [8]. We use Option-GAIL to evaluate the effect of meta-learning in DMIL. **MIL**: a transformer-based meta imitation learning method [40]. We use MIL to evaluate the effect of hierarchical structures. **MLSH**: the meta-learning shared hierarchies method [45] that relearns the high-level network in every new task. We use MLSH to evaluate the effect of fine-tuning (rather than relearning) the high-level network in new tasks. **PEMIRL**: a contextual meta inverse RL method which transfers the reward function in the new tasks [42]. We use PEMIRL to show DMIL can transfer to new tasks that have significantly different reward functions.

The second one is the Kitchen environment of the D4RL benchmark [18], which contains five different sub-tasks in the same kitchen environment. The accomplishment of each episode requires sequentially completions of four specific sub-tasks, as shown in fig. 9. We use an open demonstration dataset [48] to train our method. During training, we exclude interactions with selected objects and at test time we use demonstrations that involve manipulating the excluded object to make them unseen tasks. We choose this environment to show DMIL can be used in long-horizon tasks. We use two approaches for comparison in this experiment: **SPiRL**: an extension of the skill extraction methods to imitation learning over skill space [37]; **FIST**: an algorithm that extracts skills from offline data with an inverse dynamics model and a distance function [36].

We use fully-connected neuron networks for both the high-level network and sub-skills. More details of experiments can be found in appendix F.

## 5.2 Results

Table 1 shows success rates of different methods in ML10 and ML45 suites with sub-skill number $K = 3$. We perform 1-shot and 3-shot experiments respectively to show the progressive few-shot performance of different methods. DMIL achieves the best results in ML10 testing suite and ML45 training and testing suites. OptionGAIL achieves high success rates in both ML10 and ML45 training suites. These results show the adequate capacity of hierarchical structures to fit potential multi-modal behaviors in multi-task demonstrations. MIL achieves comparable results for all meta-testing tasks but is worse than DMIL. This shows the necessity of meta-learning processes. Compared to them, PEMIRL and MLSH are mediocre among all suites. This comes from that the reward functions across different tasks are difficult to transfer with few shot demonstrations, and the relearned high-level network of MLSH damages previously learned knowledge. We also illustrate t-sne results of these methods in fig. 4(a) to further analyze them in appendix E.2.

Table 2 shows the rewards of different methods on four unseen tasks in the Kitchen environment. *FIST-no-FT* refers to a variant of FIST that does not use future state conditioning, which makes the

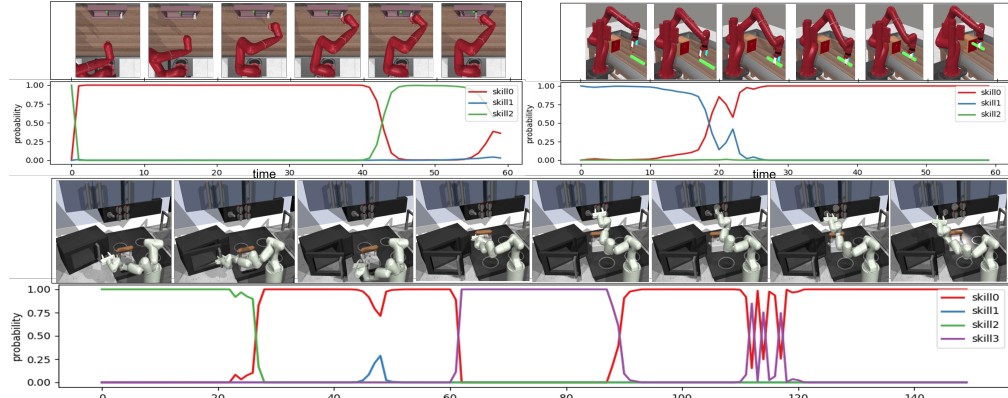

Figure 3: The iterative meta-learning process of DMIL at each iteration. Left: the supervision of high-level network (sub-skill category) comes from the most accurate sub-skill. Right: the sub-skill updated at current step is determined by the fine-tuned high-level network.

comparison fairer. DMIL achieves higher rewards on two out of four tasks and comparable results on the other two tasks, which exhibits the effectiveness of the bi-level meta-training procedure. The poor performance of DMIL on the first task may come from the choice of skill number $K$ or from low-quality demonstrations. We perform ablation studies on $K$ in the next section.

Fig. 3 shows curves of sub-skill probabilities along time of two tasks *window-close* and *peg-insert-side* of Meta-world and the *microwave-kettle-top burner-light* task in Kitchen environment. We can see the activation of sub-skills shows a strong correlation to different stages of tasks. In first two tasks, $\pi_{\theta_{l_0}}$ activates when the robot is closing to something, $\pi_{\theta_{l_1}}$ activates when the robot is picking up something, and $\pi_{\theta_{l_2}}$ activates when the robot is manipulating something. In the third task, $\pi_{\theta_{l_2}}$ activates when the robot is manipulating the microwave, $\pi_{\theta_{l_0}}$ activates when the robot is manipulating the kettle or the light switch, and $\pi_{\theta_{l_3}}$ activates when the robot is manipulating the burner switch. This shows that DMIL learns semantic sub-skills from unsegmented multi-task demonstrations.

### 5.3  Ablation Studies

In this part, we perform ablation studies on different skill numbers $K$. Due to the limited space, we put ablations on different fine-tuning steps, meta-training processes, continuity constraints, and hard/soft EM choices in appendix E.

**Effect of different skill number** $K$: Table 3 shows the effect of different sub-skill number $K$ in Meta-world experiments. We can see that a larger $K$ can lead to higher success rates on meta-training tasks, but a smaller $K$ can lead to better results on meta-testing tasks. This tells us that an excessive number of sub-skills may result in over-fitting on training data, and a smaller $K$ can play the role of regularization. In Kitchen experiments, we can see similar phenomenons in Table 5. It is worth noting in both environments, we did not encounter collapse problems, i.e., every sub-skill gets well-trained even when $K = 8$ in kitchen environment or $K = 10$ in Meta-world environments. This is because more sub-skills can help the whole structure get lower loss in the meta-training stage. However, in our supplementary videos, we can see that sub-skills trained with a large $K$ (for instance, $K = 10$ in Meta-world environments) are not as semantic as sub-skills trained by a small $K$ (for instance, $K = 3$ in Meta-world environments) during the execution of a task.

Table 3: Success rates of different sub-skill number in Meta-world environments.

|   | ML10 | | | | ML45 | | | |
|---|---|---|---|---|---|---|---|---|
|   | Meta-training | | Meta-testing | | Meta-training | | Meta-testing | |
| K | 1-shot | 3-shot | 1-shot | 3-shot | 1-shot | 3-shot | 1-shot | 3-shot |
| 2 | 0.76 | 0.955 | 0.32 | **0.72** | 0.563 | 0.818 | **0.44** | **0.67** |
| 3 | 0.775 | 0.949 | 0.396 | 0.71 | 0.59 | 0.859 | 0.376 | 0.64 |
| 5 | 0.795 | 0.94 | **0.52** | 0.57 | 0.713 | 0.92 | 0.21 | 0.48 |
| 10 | **0.8** | **0.975** | 0.38 | 0.62 | **0.736** | **0.931** | 0.34 | 0.64 |

## 6  Limitations

The limitations of DMIL come from several aspects, and future works can seek meaningful extensions in these perspectives. Firstly, DMIL models all tasks as bi-level structures. However, in real-world situations, tasks may be multi-level structures. One can extend DMIL to multi-level hierarchical structures like done in recent works [49]. Secondly, DMIL does not capture temporal information in demonstrations. Future state conditioning in Hakhamaneshi et al. [36] seems an effective tool to improve few-shot imitation learning performance in long-horizon tasks such as in the Kitchen environments. Future works can employ temporal modules such as transformer [50] as the high-level network of DMIL to improve its performance.

**Acknowledgments**

We would like to thank Tianren Zhang and Haichuan Gao for their insightful comments of the whole work, and Qualcomm China WRD UR Program. This work was supported by the National Natural Science Foundation of China 62176133.

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
