# OpenReview forum: "Transferring Hierarchical Structures with Dual Meta Imitation Learning"
_robot-learning.org/CoRL/2022/Conference — CoRL 2022 Poster_

### Official Review · Reviewer_aty5 · 2022-07-21

**Originality:** Good
**Technical Quality:** Good
**Clarity Of Presentation:** Very Good
**Impact:** 4

**Recommendation:**

Weak Accept: I recommend accepting the paper, but will not argue for my recommendation if the majority of other reviewers have a different opinion.

**Summary:**

This paper proposes a new hierarchical imitation learning structure based on MAML to learn modular sub-skills and then transfer them to new tasks. Experiments on meta-world tasks and the Kitchen environment show the proposed method outperforms previous skill-based imitation learning methods.

**Issues:**

Please refer to the pros and cons part.

I have one major concern about Equation 2 which matters whether the low level can learn meaningful sub-skills. Since all networks are initialized randomly, so why low-level policies could predict different actions given a state $s$? Why should the closest action be the supervised signal (ground truth?) to the high-level network?

Minors,
Line 173 ‘freezed’->’frozen’


**Quality Of The Limitations Section:**

Additional details required

**Reviewer Expertise:**

4: The reviewer is confident but not absolutely certain that the evaluation is correct

**Robotics Focus:**

Highly relevant to robotics but no hardware experiments

**Strengths And Weaknesses:**

Strengths:

1) This paper is well-written and easy to follow.
2) The proposed method is verified both theoretically and empirically.
3) Extensive experimental results and ablation studies show the advantage of the proposed method and the contribution of each component.

Weaknesses:

1) The structure of the paper could be improved to cover more technical details, like the details of auxiliary loss and corresponding experimental results.

2) Some references to the appendix are not clear enough, like in line 193, ‘Detailed information can be found in C.’ -> ‘Detailed information can be found in Appendix C.’ Similar in Line 205.


**Summary Of Recommendation:**

This paper clearly presents the proposed method and provides theoretical analysis as well as empirical analysis. I did not check the technical details of the theoretical analysis carefully. But based on my current understanding and if the authors address my concern, I would vote for acceptance of this paper.

---

> ### Author Response · Authors · 2022-08-17
> **Response to Reviewer aty5**
>
> Dear reviewer aty5,
>
> We sincerely appreciate your valuable and insightful comments. We address your questions in detail below.
>
> ### Q1: The structure of the paper could be improved to cover more technical details, like the details of auxiliary loss and corresponding experimental results.
>
> A1: Thanks for your suggestion. We will improve the structure of our paper in the final version, which will have one more page for an better exhibition of more results.
>
> ### Q2: Some references to the appendix are not clear enough.
>
> A2: Thanks very much for your suggestion. We will rewrite these parts in the final version.
>
> ### Q3: Since all networks are initialized randomly, so why low-level policies could predict different actions given a state S? Why should the closest action be the supervised signal (ground truth?) to the high-level network?
>
> A3: Thanks for your question. This is the principle of EM algorithm, which is an unsupervised clustering method. Although each sub-skill is randomly initialized, by iteratively updating the hierarchical structure according to the maximum likelihood principle, we can get a set of converged groups that represent different policies, and the specific forms of the maximum likelihood principle in our paper correspond to Equation (2) and (4), in cross-entropy loss and mean-square loss forms. For the detailed proof, please refer to Appendix D.

---

### Official Review · Reviewer_dgTJ · 2022-08-01

**Originality:** Good
**Technical Quality:** Very Good
**Clarity Of Presentation:** Very Good
**Impact:** 3

**Recommendation:**

Weak Accept: I recommend accepting the paper, but will not argue for my recommendation if the majority of other reviewers have a different opinion.

**Summary:**

This paper tackles the following two problems (simultaneously):
- Given a dataset of demonstrations from many different tasks, learn a set of hierarchical policies that solve these tasks well.
- Learn the hierarchical policy above in such a way that when given a new task, you can quickly adapt both the high-level and low-level policies to this task.

The former problem has been explored in the hierarchical imitation learning (HIL) literature, while the latter problem has been well-explored in the meta-imitation learning (MIL) literature. The main contribution of this paper is to address both of these problems at the same time. The proposed method combines an EM-style algorithm (popular in HIL) with MAML (popular in MIL). Experiments are performed on simulated robotics tasks, and show results that are decent (method works), but not stellar (comparable performance to competing methods on 2 out of the 3 benchmarks).


**Issues:**

Please take a look at the weaknesses, and use them to either update the paper or expand the limitations section.

**Quality Of The Limitations Section:**

Additional details required

**Reviewer Expertise:**

4: The reviewer is confident but not absolutely certain that the evaluation is correct

**Robotics Focus:**

Highly relevant to robotics but no hardware experiments

**Strengths And Weaknesses:**

Strengths
- The paper tackles an important problem (transferring subskills across tasks), that I believe is somewhat under-studied in the hierarchical imitation learning literature.
- The paper is well-written.
- Authors demonstrate results on manipulation tasks, which is nice to see (a lot of hierarchical learning papers end up focusing on locomotion / navigation tasks, where the hierarchical structures are easier find).

Weaknesses
- The link provided for code and demonstrations points to an expired repository: https://anonymous.4open.science/r/DMIL
- “DMIL achieves higher rewards on two out of four tasks and comparable results on the other two tasks, which exhibits the effectiveness of the bi-level meta-training procedure.” – I am not sure if this is a fair assessment. The performance for DMIL is only higher for one task (the third task), and the results are comparable for the rest of the three tasks. The confidence intervals appear to be quite large for both methods. Similarly, for ML10, DMIL results are comparable to vanilla MIL. Overall, I would say that the empirical performance is decent (the method works), but not stellar (it’s only significantly better for ML45, one out of the 3 benchmarks studied in the paper).
- Most meta-world tasks are fairly short (pick + place tasks, or turning a lever), so it’s unclear to me if a hierarchical method is actually needed for these tasks. As shown in the results K=2 works best for meta-world tasks. When more options are assigned, they typically go unused (Figure 2). The Kitchen tasks are longer, but the performance when compared against competing methods is not particularly stellar.
- The meta-imitation problem setting is well-suited to real world learning (since it can be done entirely offline), so it's a bit disappointing that the paper does not have any real robot results. Perhaps the proposed meta-learning algorithm has stability issues when working under real world constraints such as visual observations? If that is the case, it should be mentioned in the limitation sections.


**Summary Of Recommendation:**

Overall, while the paper has weaknesses, I think it's technically solid, and targets an important problem, so I am leaning towards acceptance. Note that I did not check the theoretical convergence results (the intuitive derivation combining EM and MAML made sense to me).

---

> ### Author Response · Authors · 2022-08-17
> **Response to Reviewer dgTJ**
>
> Dear reviewer dgTJ,
>
> We sincerely appreciate your valuable and insightful comments. We are sorry that the code link expires, and this is the new link: https://anonymous.4open.science/r/DMIL_new. We address your questions in detail below.
>
> ### Q1: The empirical performance is decent (the method works), but not stellar (it’s only significantly better for ML45, one out of the 3 benchmarks studied in the paper).
>
> A1: Yes, DMIL does not achieve a landslide superiority on all suites and all tasks, but we think that we cannot say DMIL is just **decent**. In the Kitchen environments, DMIL achieves better results on half of the tasks, and comparable results on the other half, thus we could say that DMIL has similar performance to SOTA methods. For Meta-world tasks, DMIL is no better than baselines on ML10 training tasks but is better than them on ML10 testing tasks, ML45 training tasks, and ML45 testing tasks (especially on testing tasks). This shows that in a small set of tasks, some previous methods can also perform well. But when the task number increases, the adaptive hierarchical structure begins to show its advantages.
>
> ### Q2: It’s unclear to me if a hierarchical method is actually needed for meta-world tasks.
>
> A2: Thanks for your question. In our experiments, the MIL baseline is a non-hierarchical MAML imitation learning algorithm, and we show in Table 1 that MIL does achieve competitive results on some suites (especially on ML10 tasks). This may show that one monolithic but complex network (e.g. a transformer-based policy network) may be enough for ML10 tasks. However, our experiments in ML45 tasks show that when the task number increases, the hierarchical structure starts to show its usefulness and advantage over flat architectures.
>
> ### Q3: It's a bit disappointing that the paper does not have any real robot results.
>
> A3: Thanks for this question. Yes, we should add a real robot experiment for DMIL considering the theme of CORL. This is absolutely the best choice. However, according to [1] [2] [3], MAML-based and hierarchical-based one-shot imitation learning algorithms both perform well on real-world tasks with regular CNN networks to process the image inputs, thus we have reasons to believe that DMIL will have promising results on real-world robots. So in this paper, we think using simulation environments is enough to show the effectiveness of DMIL as done in other related works [4].
>
> ------
> [1] C. Finn, T. Yu, T. Zhang, P. Abbeel, and S. Levine. One-shot visual imitation learning via meta-learning. In 1st Annual Conference on Robot Learning, 2017.
>
> [2] T. Yu, P. Abbeel, S. Levine, and C. Finn. One-shot hierarchical imitation learning of compound visuomotor tasks. arXiv preprint arXiv:1810.11043, 2018.
>
> [3] T. Yu, C. Finn, S. Dasari, A. Xie, T. Zhang, P. Abbeel, and S. Levine. One-shot imitation from observing humans via domain-adaptive meta-learning. In Robotics: Science and Systems XIV, 2018.
>
> [4] L. Yu, T. Yu, C. Finn, and S. Ermon. Meta-inverse reinforcement learning with probabilistic context variables. In Advances in Neural Information Processing Systems, 2019.

---

> > ### Comment · Reviewer_dgTJ · 2022-08-26
> > **Response acknowledgment**
> >
> > Thank you for your response.

---

### Official Review · Reviewer_ZhJq · 2022-08-01

**Originality:** Good
**Technical Quality:** Good
**Clarity Of Presentation:** Fair
**Impact:** 3

**Recommendation:**

Weak Accept: I recommend accepting the paper, but will not argue for my recommendation if the majority of other reviewers have a different opinion.

**Summary:**

Use MAML-style gradient-based meta-learning to learn a hierarchical structure from imitation learning. This requires an inner loop that fine-tunes the controller and the sub-skills and an outer loop that updates the meta-gradients. The order is from inner to outer, where the first step decides the assignment of sub-skills and assigns loss based on the behavior cloning loss of that sub-skill (log-likelihood). Then the second inner loop updates the sub-skills based on the assignments given by the controller to minimize behavior cloning. The outer loop then updates the meta-gradients. This work also demonstrates that the inner and outer gradients are equivalent to E and M steps for coordinate descent, implying that the algorithm will coverge to a local minima.

**Issues:**

Most of the issues are in the weaknesses section, but mostly the paper could benefit from having some more extensive experiments related to determining why such little variation is needed in the number of skills. In addition, the writing was overall difficult to follow and could be improved by some editing.

**Quality Of The Limitations Section:**

Limitations are not well addressed

**Reviewer Expertise:**

3: The reviewer is fairly confident that the evaluation is correct

**Robotics Focus:**

Relevant but unlikely to deploy to hardware in near future

**Strengths And Weaknesses:**

Strengths:
This work provided an interesting algorithm, clever implementation and good experiments

Weaknesses:
The writing is difficult to follow, using ill-defined terms like 23 "plenty", or 32 "come from that the" or "may conflict with descrpant situations". As it is, the second paragraph in the intriduction is difficult to understand and after several readings, it is still hard to determine exactly what it is saying. The third paragraph has the same issue, with statements like "the same question applies to the high-level network", when referring to what high-level network that low level sub-skills should choose, which does not appear to make sense. It was difficult to gain more insight from the introduction than what was stated in the abstract.

It is not entirely clear from the methods section how the meta-gradients actually work, and considering that is the main point of the algorithm (the rest is similar in concept to the EM algorithms that generally get used), that is somewhat disappointing. The proof is difficult to follow and thus to verify, though it appears intuitive enough: by making it equivalent to the EM algorithm, the algorithm will converge.

Overall, the choice of meta-world for tasks is good and the performance appears to be quite effective, especially with 45 tasks. However, it does seem a little bit suspect that 2 sub-skill performed best, considering that it seems like that implies that two general behavior patterns solve 45 different tasks. Would 1 task have performed even better? Either way, it seems like the MAML component may be where the performance comes from, rather than the skill learning, and some non-hierarchical change in architecture resulted in better training.

**Summary Of Recommendation:**

This paper provides an insightful algorithm that seems like it has the potential to be built upon and evaluated in a classic benchmark with good results. While some of the specifics of the algorithm, such as the way that the number of sub-skills appears not to scale, cast some doubt on the algorithm or the evaluation, for the most part, it appears to be a valuable contribution. I was not particularly impressed by the proof, but that might be more related to my background than to any limitations with it.

---

> ### Author Response · Authors · 2022-08-17
> **Response to Reviewer ZhJq**
>
> Dear reviewer ZhJq,
>
> We sincerely appreciate your valuable and insightful comments. We address your questions in detail below.
>
> ### Q1: The writing is difficult to follow. The second and third paragraph in the introduction is difficult to understand.
>
> A1: We are sorry that we did not make our paper clear to you. We will modify the ill-defined terms you mentioned in our paper in the final version. We hope the following explanation can alleviate your confusion.
>
> The second paragraph illustrates the necessity of dual meta-learning in few-shot imitation learning tasks, and we are mainly talking about the examples in Fig. 1(a). In this figure, the high-level network needs to adapt to new requirements since the left task needs to call the “pull” skill to open the drawer while the right task needs to call the “push” skill to close the drawer. The sub-skill network of the left side is a general form of “pick-place” skill, while on the right side, this sub-skill needs to adapt to different specific forms in new tasks (picking up different objects may need different picking motions).
>
> The third paragraph points out the difficulties to use MAML for our purpose. The main difficulty is to determine the correct order to adapt high-level and low-level networks to ensure convergence. We are sorry that the writing of this paragraph may be a little verbose to confuse you.
>
> ### Q2: How do the meta-gradients actually work?
>
> A2: Thanks for your question. Meta-gradients correspond to the *HO* and *LO* steps of DMIL. In brief, we use the inner-adapted parameters of dual-level networks to get the meta-gradients for them. It is important that the meta-updating of dual-level networks must be done at the same time after getting the meta-gradients of a batch of tasks, rather than sequentially, to ensure convergence.
>
> ### Q3: It seems like the MAML component may be where the performance comes from, rather than the skill learning, and some non-hierarchical changes in architecture resulted in better training.
>
> A1: Thanks for your question. In our experiments, the MIL baseline is a non-hierarchical MAML imitation learning algorithm, and we show in Table 1 that MIL does achieve competitive results on some suites (especially on ML10 tasks). This may show that one monolithic but complex network (e.g. a transformer-based policy network) may be enough for ML10 tasks. However, our experiments in ML45 tasks show that when the task number increases, the hierarchical structure starts to show its usefulness and advantage over flat architectures.

---

> > ### Comment · Reviewer_ZhJq · 2022-08-26
> > **Response to Authors**
> >
> > Thank you for your detailed responses, they helped clarify questions related to the paper and are much appreciated

---

### Official Review · Reviewer_iytY · 2022-08-01

**Originality:** Good
**Technical Quality:** Very Good
**Clarity Of Presentation:** Very Good
**Impact:** 4

**Recommendation:**

Weak Accept: I recommend accepting the paper, but will not argue for my recommendation if the majority of other reviewers have a different opinion.

**Summary:**

Paper proposes a new method for meta-learning hierarchical imitation learning, provides theoretical analysis of the algorithm showing its convergence, and demonstrates SOTA results on the meta-world benchmark and competitive results on the D4RL kitchen environments.

**Issues:**

See weaknesses section.

**Quality Of The Limitations Section:**

Limitations are addressed clearly

**Reviewer Expertise:**

2: The reviewer is willing to defend the evaluation, but it is quite likely that the reviewer did not understand central parts of the paper

**Robotics Focus:**

Relevant but unlikely to deploy to hardware in near future

**Strengths And Weaknesses:**

Strengths
- Nicely written and easy to understand
- Provides both theoretical analysis of the algorithm showing its convergence, and empirical results showing its benefits to prior methods
- Includes exhaustive ablation studies showing the importance of each component of the algorithm

Weaknesses
- The proposed method is rather complex, which may make it difficult to be deployed on real systems.
- It would be nice to include some intuitive discussion on why *both* the high level and low level components need to adapt to a new task? Since many different subskills can be represented with the low level component, it seems like maybe only the high level may need to adapt given a new task, by picking the right sequence of subskills?

**Summary Of Recommendation:**

This paper provides an interesting new method and demonstrated superior empirical results compared to prior work.

---

> ### Author Response · Authors · 2022-08-17
> **Response to Reviewer iytY**
>
> Dear reviewer iytY,
>
> We sincerely appreciate your valuable and insightful comments. We address your questions in detail below.
>
> ### Q1: The proposed method is rather complex, which may make it difficult to be deployed on real systems.
>
> A1: Our method DMIL is a fully offline meta imitation learning algorithm, which theoretically can be conveniently deployed to real robots, given enough and diverse demonstrations on different tasks. Testing DMIL on real-world robots with high-dimensional raw image inputs can be a meaningful future work, but according to [1] [2] [3], MAML-based one-shot imitation learning algorithms perform well on real-world tasks with common CNN networks to process the image inputs. Thus in this paper, we think using simulation environments is enough to show the effectiveness of DMIL as done in other related works [4].
>
> ### Q2: It would be nice to include some intuitive discussion on why both the high level and low-level components need to adapt to a new task. Since many different subskills can be represented with the low-level component, it seems like maybe only the high level may need to adapt given a new task, by picking the right sequence of subskills?
>
> A2: Thanks for your question! You mentioned that *many different subskills can be represented with the low-level component*. This is the viewpoint of **multi-task learning**, which supposes one sub-skill network can handle different kinds of sub-skills without adaptation to new tasks. However, in the **meta-learning** perspective, although we can learn good initial network parameters for each sub-skill, it still needs to be adapted to new tasks for better performance based on the requirements of new tasks.
>
> For example, consider a sub-skill network that represents the skill “pushing”. Note that “pushing a cube” and “pushing a football” requires different specific motions, since the football may roll away during pushing. By adapting sub-skill networks with new task few-shot demonstrations, each sub-skill can be fine-tuned to meet the specific requirements of new tasks.
>
>
> ------
> [1] C. Finn, T. Yu, T. Zhang, P. Abbeel, and S. Levine. One-shot visual imitation learning via meta-learning. In 1st Annual Conference on Robot Learning, 2017.
>
> [2] T. Yu, P. Abbeel, S. Levine, and C. Finn. One-shot hierarchical imitation learning of compound visuomotor tasks. arXiv preprint arXiv:1810.11043, 2018.
>
> [3] T. Yu, C. Finn, S. Dasari, A. Xie, T. Zhang, P. Abbeel, and S. Levine. One-shot imitation from observing humans via domain-adaptive meta-learning. In Robotics: Science and Systems XIV, 2018.
>
> [4] L. Yu, T. Yu, C. Finn, and S. Ermon. Meta-inverse reinforcement learning with probabilistic context variables. In Advances in Neural Information Processing Systems, 2019.

---

> > ### Comment · Reviewer_iytY · 2022-08-26
> > **Reply to authors**
> >
> > Thank you for your explanations.

---

### Meta-Review · Area_Chair_wjsc · 2022-08-14

**Recommendation:** Accept (Poster)
**Confidence:** 4

**Metareview:**

Phase 1:

Strengths:
The paper is overall well and intuitively written. It includes convincing theoretical and empirical results and detailed ablation studies. Most reviewers appreciate the sound technical contributions and simulation-based experiments.

Weaknesses:
A big limitation is the absence of real robot experiments and some of the reviewers doubt future real-world application given complexity. Further, best results on metaworld 45 point towards the minimal number of sub-skills working best (k=2) and the reviewers would like to see this point discussed in further detail. Particular in light of the discussion of limitations which is called out across all reviewers as a point of required improvements. Further minor points target the slight overemphasis on potential gains and limited success in domains other than ML45.

Phase 2:

The feedback has been overall positive with 4 weak accepts but also includes some criticism (limitations, specific gains with smallest number of skills). I agree with the reviewers and recommend acceptance. Please take the remaining points from the review process seriously and follow up with improvements on open points and promised changes.

**Best Paper Nomination:**

No